# Formation and Long-Term Culture of hiPSC-Derived Sensory Nerve Organoids Using Microfluidic Devices

**DOI:** 10.3390/bioengineering11080794

**Published:** 2024-08-05

**Authors:** Takuma Ogawa, Souichi Yamada, Shuetsu Fukushi, Yuya Imai, Jiro Kawada, Kazutaka Ikeda, Seii Ohka, Shohei Kaneda

**Affiliations:** 1Mechanical Engineering Program, Graduate School of Engineering, Kogakuin University, 1-24-2 Nishishinjuku, Shinjuku-ku, Tokyo 163-8677, Japan; 2Department of Virology I, National Institute of Infectious Diseases, 1-23-1 Toyama, Shinjuku-ku, Tokyo 162-8640, Japan; 3Jiksak Bioengineering, Inc., 3-25-16 Tonomachi, Kawasaki-ku, Kawasaki 210-0821, Kanagawa, Japan; 4Addictive Substance Project, Tokyo Metropolitan Institute of Medical Science, 2-1-6 Kamikitazawa, Setagaya-ku, Tokyo 156-8506, Japanohka-si@igakuken.or.jp (S.O.); 5Department of Neuropsychopharmacology, National Institute of Mental Health, National Center of Neurology and Psychiatry, 4-1-1 Ogawahigashi-cho, Kodaira, Tokyo 187-8553, Japan

**Keywords:** sensory nerve organoid, sensory neuron, iPSC, microfluidic device, microchannel, bioengineered nociceptive sensors

## Abstract

Although methods for generating human induced pluripotent stem cell (hiPSC)-derived motor nerve organoids are well established, those for sensory nerve organoids are not. Therefore, this study investigated the feasibility of generating sensory nerve organoids composed of hiPSC-derived sensory neurons using a microfluidic approach. Notably, sensory neuronal axons from neurospheres containing 100,000 cells were unidirectionally elongated to form sensory nerve organoids over 6 mm long axon bundles within 14 days using I-shaped microchannels in microfluidic devices composed of polydimethylsiloxane (PDMS) chips and glass substrates. Additionally, the organoids were successfully cultured for more than 60 days by exchanging the culture medium. The percentage of nuclei located in the distal part of the axon bundles (the region 3−6 mm from the entrance of the microchannel) compared to the total number of cells in the neurosphere was 0.005% for live cells and 0.008% for dead cells. Molecular characterization confirmed the presence of the sensory neuron marker ISL LIM homeobox 1 (ISL1) and the capsaicin receptor transient receptor potential vanilloid 1 (TRPV1). Moreover, capsaicin stimulation activated TRPV1 in organoids, as evidenced by significant calcium ion influx. Conclusively, this study demonstrated the feasibility of long-term organoid culture and the potential applications of sensory nerve organoids in bioengineered nociceptive sensors.

## 1. Introduction

The somatic and autonomic nervous systems constitute the peripheral nervous system, with the somatic nervous system consisting of sensory and motor nerves. Somatic neurons derived from human induced pluripotent stem cells (hiPSCs) have been used to generate neuronal disease models [1,2,3]. For example, sensory neurons derived from hiPSCs have been used to generate cell models of inherited erythromelalgia (IEM) [4,5], chronic insensitivity to pain (CIP) [6], and small fiber neuropathy (SFN) [7]. Additionally, motor neurons derived from hiPSCs have been used to establish models of amyotrophic lateral sclerosis (ALS) [8,9] and spinal muscular atrophy (SMA) [10,11].

Recent advances in microfluidic neuronal cell culture have allowed the formation of bundle-shaped structures of motor neuronal axons, referred to as motor nerve organoids [12,13,14], from hiPSCs for in vitro experiments. Notably, motor neuronal axons from neurospheres (a spherical aggregation of neuronal cells) are unidirectionally elongated along the microchannel and spontaneously organized into a fascicle (i.e., axon bundle), using an I-shaped microchannel in microfluidic devices. Importantly, the macroscopic size of the motor nerve organoids allows their manipulation using tweezers, making it easier for the organoids to be divided into axon bundle and neurosphere fractions with a knife and allowing for easy and separate collection with a micropipette. Axon bundles of motor nerve organoids are emerging as a useful tool for axon-enriched gene analyses, especially for pathological ALS studies [15,16], and are useful transplantation materials for nerve regeneration after peripheral nerve injury [17].

Although motor nerve organoids offer the aforementioned attractive applications, studies on the formation of sensory nerve organoids are limited because of the difficulties of the long-term culturing of functional long axon bundles. In this study, we developed a microfluidic approach for the formation of sensory nerve organoids composed of sensory neurons derived from hiPSCs. Additionally, we elucidated the duration required for the formation of sensory nerve organoids through axon elongation in I-shaped microchannels, the long-term culturability of the organoids, and numbers of nuclei in the axon bundles of the organoids. Moreover, we detected the expression of the capsaicin receptor transient receptor potential vanilloid 1 (TRPV1) [18] on the organoids and investigated the functionality of TRPV1 in response to capsaicin stimulation.

## 2. Materials and Methods

### 2.1. Device Design and Fabrication

In this study, we designed and fabricated a microfluidic device composed of a polydimethylsiloxane (PDMS) chip and a glass substrate (Figure 1a). The length and narrowest width of the microchannel were 6 mm and 150 μm, respectively. The neurosphere chamber and axon terminal chamber located at both ends of the microchannel had an access hole of 1.5 mm in diameter for the introduction of the culture medium into the microchannel. Four microchannels were present in a single device (Figure 1b). Notably, the PDMS chip was fabricated as previously described [13]. Thereafter, the fabricated PDMS chip and glass substrate, in a glass-bottom dish (35 mm diameter, 3910-035-IN, AGC Techno Glass, Shizuoka, Japan), were irreversibly bonded to each other via oxygen plasma treatment [19] using a reactive-ion etching machine (RIE-10NR, SAMCO, Kyoto, Japan). Importantly, the glass-bottom dish served as a 3 mL reservoir for the preservation of the nerve organoid culture medium, whereas in previous reports [12,13], the reservoirs were fabricated on the PDMS chips. This improvement simplifies the fabrication process of the device and suppresses microbial contamination during organoid culture.

### 2.2. Cell Culture for Sensory Nerve Organoid Formation and Characterization

Except for sensory neuron induction culture medium, the protocol for sensory nerve organoid formation using the microfluidic device was similar to the previously described protocol for motor nerve organoids [13]. Briefly, a human iPS cell line (409B2) [20] was seeded in a vitronectin-coated 6-well plate (Thermo Fisher Scientific, Waltham, MA, USA) containing xeno-free and feeder-free E8 medium (Essential 8™ Flex Medium, Thermo Fisher Scientific). For sensory neuron induction, cells were treated with a combination of five small-molecule pathway inhibitors, including LDN193189 hydrochloride (Sigma Aldrich, St. Louis, MO, USA), SB431542 (Sigma Aldrich), CHIR9902 (ChemScene, Monmouth Junction, NJ, USA), SU5402 (Sigma Aldrich), and DAPT (Sigma Aldrich), as described by Chambers et al. [21]. After induction, the differentiated cells (100,000 cells per 200 μL) were seeded in N2B27 medium [13] containing 10 μM of Y-27632 [22] (FUJIFILM Wako Pure Chemical Corp., Osaka, Japan) in a V-bottom well of a 96-well plate with an ultra-low-cell-adhesion surface (Sumitomo Bakelite, Tokyo, Japan) [23] on day 2 to form a neurosphere. The medium was replaced with N2B27 medium without Y-27632 on day 1. Thereafter, the neurospheres were transferred to the devices and cultured in 3–4 mL of N2B27 medium on day 0 to generate axonal fascicles. The medium was changed every 3–4 days. The device was sterilized via UV exposure, and the microchannel surfaces were coated for 24 h at 4 °C with gel derived from mouse Engelbreth–Holm–Swarm sarcoma (ECM gel, Sigma Aldrich) diluted 80 times with N2B27 medium to enhance the adhesion of the neurospheres before transferring them to the device. Before filling the microchannels with the coating solution, the device was degassed for 40 min using a vacuum container (VCP-15L, AS ONE, Osaka, Japan) to evacuate air bubbles from the microchannels that may have formed during the process of filling the coating solution [24]. Notably, the differentiated cells were cryopreserved using CryoStor^®^ CS10 (BioLife Solutions, Bothell, WA, USA), and the dilution ratio of the coating solution was lower than that used for motor nerve organoid culture [13].

An inverted fluorescence microscope (BZ-X810, Keyence, Osaka, Japan) equipped with a CCD camera and a motorized XYZ stagewas used to obtain bright-field and fluorescence images of sensory nerve organoids cultured in the microchannels. The sensory nerve organoids were stained with calcein-AM (Dojindo Laboratories, Kumamoto, Japan), propidium iodide (PI) (Dojindo Laboratories), and Hoechst 33342 (Dojindo Laboratories) to identify the positions of the nuclei in the axon bundles of the organoids. The cells and the nuclei of the nerve organoids were labeled using calcein-AM, PI, and Hoechst 33342 in microchannels without cell fixation, followed by observation using the fluorescence microscope. For molecular characterization, the sensory nerve organoids were labeled with anti-tubulin β3 (TUBB3/TUJ1) mouse monoclonal antibody (MMS-435P, BioLegend, CA, USA), anti-ISL LIM homeobox 1 (ISL1) rabbit polyclonal antibodies (GTX102807, GeneTex, TX, USA), or anti-TRPV1 rabbit polyclonal antibodies (NBP1-97417, Novus Biologicals, CO, USA), followed by incubation with Alexa Fluor-conjugated anti-mouse or anti-rabbit polyclonal secondary antibodies (ab150109, ab150068, Abcam, Waltham, MA, USA). After peeling off the entire organoid from the microchannels using a flow of 200 μL of medium generated with a pipette from the axon terminal chamber, the organoids were fixed and permeabilized with 4% paraformaldehyde phosphate-buffer solution (FUJIFILM Wako Pure Chemical Corp.) and 0.5% Triton X-100 (Sigma Aldrich), respectively. Thereafter, the organoids were stained with primary and secondary antibodies, and the nuclei were labeled with Hoechst 33342. To validate the functionality of TRPV1 expressed on the organoids, TRPV1 was stimulated with capsaicin (FUJIFILM Wako Pure Chemical Corporation, Osaka, Japan) dissolved in DMSO at 100 μM. After removing the culture medium from the reservoir of the microfluidic device, 3 mL of capsaicin solution or DMSO was poured into the reservoir to cover the entire surface of the PDMS chip. Calcium ion (Ca^2+^) influx following capsaicin stimulation was visualized using Fluo 4-AM [18] (Dojindo), a fluorescence probe for intercellular calcium ion influx, under the fluorescence microscope.

## 3. Results

To investigate whether the I-shaped microchannels can maintain hiPSC-derived sensory neurons with bundles for a long period, the neurospheres containing 100,000 cells each were seeded into one of the neurosphere ports, with one neurosphere per port, on day 0. Axons of hiPSC-derived sensory neurons were unidirectionally elongated into microchannels and formed a bundle of at least 6 mm in length 14 days after seeding the neurospheres (Figure 2a). The bundle shape was maintained for over 39 days without shrinkage (Figure 2b). Neurospheres with axonal bundles were maintained for over 62 days in the microchannels without apparent shrinkage. During the culture period, the diameter of the axon bundle increased in size (diameter at 3 mm from the entrance of the microchannel was 89 μm on day 30 and 110 μm on day 62, Figure 3).

Although the microchannels are narrow and shallow enough to prevent neurospheres from intruding into them, cells detached from the neurosphere can flow into the microchannel and attach to the axon bundle, or cells existing on elongating axons can be transported into the axon bundles. To investigate the existence of live cells in axon bundles, we examined whether the axon bundles contained nuclei derived from live cells. Calcein-AM fluorescence, an indicator of the interior of live cells, including live axons, demonstrated that the microchannel was filled with live axons (Figure 4a). Hoechst 33342 and PI fluorescence staining, indicators of nuclei derived from live cells and dead cells, respectively, indicate that nuclei stained with Hoechst 33342^+^/PI^−^ are derived from live cells, whereas nuclei stained with Hoechst 33342^+^/PI^+^ are derived from dead cells. Nuclei derived from live cells mainly existed in the neurosphere and partly in the axon bundle (Figure 4a). Notably, the numbers of nuclei derived from live and dead cells located in each 1 mm interval were comparable, and the sum of the average numbers of nuclei located in the 3–6 mm region was 5.0 per axon bundle for live cells and 8.2 per axon bundle for dead cells. Additionally, the percentage of nuclei located in the 3–6 mm region compared to the total number of cells in the neurosphere was 0.005% for live cells and 0.008% for dead cells (Figure 4b). Conclusively, these results suggested that few live cells reached the 3 mm region, and axon bundle contamination with live cells was almost negligible in regions exceeding 3 mm.

Fluorescence immunostaining was performed to characterize the neurospheres and axon bundles of organoids. We detected the neuronal marker tubulin β3 (TUJ1) and the transcription factor ISL1, a sensory neuron marker, in the induced sensory neurons and neurospheres prior to seeding them into the device (Appendix A). On day 28–39, ISL1 was still expressed in the neurospheres (Figure 5a), and the capsaicin receptor TRPV1, a nociceptor marker, was expressed throughout the organoids, including in the axon bundles (Figure 5b). Overall, these results indicated that organoids cultured for around 30–40 days maintained the characteristics of sensory nerve organoids, including nociceptive sensory neurons.

Furthermore, we examined the activity of TRPV1 in sensory nerve organoids cultured for 32 days following capsaicin stimulation (Figure 6). The fluorescent probe Fluo 4-AM was used to visualize intracellular calcium ion influx. Capsaicin stimulation significantly increased Fluo 4-AM fluorescence intensity in the neurosphere as well as in the axon bundle, whereas mock stimulation with DMSO did not (Figure 6a,b see Appendix A). These results suggest that capsaicin stimulates TRPV1 in sensory nerve organoids, resulting in calcium ion influx, and that sensory nerve organoids have functional TRPV1. Conclusively, sensory nerve organoids generated in microchannels can sense capsaicin via calcium ion influx.

## 4. Discussion

In this study, we generated sensory nerve organoids composed of sensory neurons derived from hiPSCs, using a microfluidic approach. Although we were not able to measure the exact length of the axons beyond 6 mm, the axon elongation rate from day 0 to day 11 was approximately 0.54 mm per day (Figure 2b), which was consistent with the elongation rate observed in motor nerve organoids (0.44 to 0.60 mm per day) in a previous study [13]. Chambers et al. [21] reported that sensory neurons were successfully cultured for 20 days after induction using a medium containing human β-nerve growth factor (NGF), brain-derived neurotrophic factor (BDNF), and glial cell-derived neurotrophic factor (GDNF). In contrast, sensory neurons associated with nerve organoids in the present study were cultured for over 60 days in a medium without the aforementioned expensive neurotrophins. Importantly, the axon elongation rate and the axon bundle diameters recorded following incubation in N2B27 medium without neurotrophins were higher than those obtained in cells incubated in N2 medium containing the aforementioned neurotrophins (Appendix A). Additionally, the long-term culturability of the sensory nerve organoids generated in this study (62 days; Figure 3) was comparable to that of motor nerve organoids in our previous study (72 days) [13]. Moreover, the increase in axon bundle diameter due to the increase in the number of associated axons suggests that the medium used in this study is suitable for long-term culture (Figure 3). Chambers et al. [21] reported that induced sensory neurons organized into ganglia-like clusters by day 30 due to increased cell–cell adhesion. Notably, we hypothesized that this increase in cell–cell adhesion results in a relative decrease in cell–matrix adhesion, which may cause the detachment of neurons from the culture substrate. Considering that the sensory nerve organoids formed in the present study, especially the axon bundles, were protected by microchannels, it could be concluded that the microchannels may be useful for preventing the detachment of organoids caused by flow during culture medium exchange.

Neurospheres containing 100,000 cells were used in this study. Notably, we confirmed that nuclei derived from both live and dead cells were mainly present in the neurosphere and partly in the axon bundle (Figure 4). Although the number of cells in axon bundles was low, the effects of nuclei present in the axon bundle on axon-enriched gene analysis [15,16] should be investigated in future studies.

Consistent with previous findings [21], ISL1 and TRPV1 were detected in the organoids (Figure 5b). Additionally, we examined the functionality of TRPV1 in the organoids following capsaicin stimulation (Figure 6). Compared with that in the neurosphere, the increase in fluorescence intensity, indicating calcium ion influx, was delayed in the axon bundle (Figure 6b). These results imply the possibilities that there was a delay in the transfer of the capsaicin solution to the axon bundle, and that the capsaicin signal was transmitted from the cell bodies to the axon bundles under the experimental conditions. According to a previous study [21], TRPV1 activation was observed in a rare subset of neuronal processes and cell bodies (1–2% of induced sensory neuron cells in monolayer culture) following capsaicin stimulation. In the present study, we were able to easily visualize TRPV1 activation in response to the capsaicin treatment due to the tremendous number of cells associated with the formed organoids and its three-dimensional structure (Figure 6a). Together, the sensory organoids in the microchannels may be highly suitable for sensing nociceptive stimuli. In a previous study [21], induced sensory neurons derived from hiPSCs expressed not only TRPV1 but also other nociceptor-specific channels and receptors, such as the pain transmission-related channels sodium voltage-gated channel alpha subunit 9 (SCN9A/Nav1.7), sodium voltage-gated channel alpha subunit 10 (SCN10A/Nav1.8), sodium voltage-gated channel alpha subunit 11 (SCN11A/Nav1.9), the pain preceptor P2X purinoceptor 3 (P2RX3), and the cold and menthol receptor transient receptor potential cation channel subfamily M member 8 (TRPM8). Collectively, these results suggest that sensory nerve organoids can be bioengineered as cell-based sensors to measure these stimuli and their transmission.

Despite these promising findings, this study had several limitations. For example, only one iPSC line (409B2) was used, indicating the need for further studies to investigate the feasibility of using other cell lines for the formation of sensory nerve organoids. Besides other iPSC lines, both the further identification of other cell types included in the organoids and further validation of the functionality of TRPV1 in the organoids cultured long-term are necessary. Additionally, the induction of sensory neuron cells and organoid formation was performed using an ECM gel derived from mouse Engelbreth–Holm–Swarm sarcoma, which prevented the establishment of xeno-free, fully defined culture conditions. Notably, the use of human recombinant laminin fragments [25,26] as alternatives to ECM gels is a promising approach. Nociceptors have two types of axons: central axons, which connect to the central nervous system, and peripheral axons, which have free nerve endings. Although we assume that the majority of neurons comprising the sensory nerve organoids were nociceptors because of the induction protocol used in this study [21], the bipolarity of the axons in the organoids was lost. To overcome this limitation, using two-branched microchannels and generating concentration gradients of axon guidance cues (e.g., NGF and BDNF) in the microchannels of each axon are effective strategies.

## 5. Conclusions

In this study, we developed a microfluidic device for generating sensory nerve organoids. Neuronal axons from neurospheres containing 100,000 cells were unidirectionally elongated to form sensory nerve organoids over 6 mm long axon bundles within 14 days, using I-shaped microchannels. The generated organoids were cultured for over 60 days without detachment by exchanging the culture medium without using expensive neurotrophins. Overall, these findings indicate that the microfluidic device was effective for the formation of functional sensory nerve organoids. Moreover, the sensory nerve organoids possess potential medical applications, including in pathological studies with axon-enriched gene analyses using RNA-seq [15,16] and axon bundle transplantation for nerve regeneration after peripheral nerve injury [17].

## Figures and Tables

**Figure 1 bioengineering-11-00794-f001:**
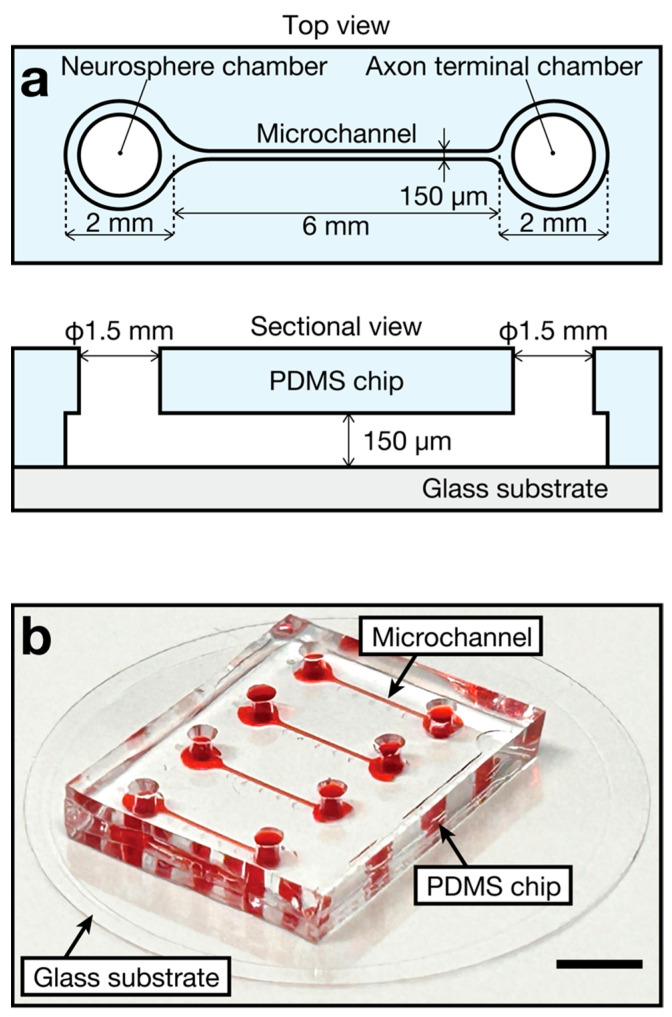
Microfluidic device for generating sensory nerve organoids. (**a**) Device design. (**b**) Photograph of the fabricated device. Scale bar: 5 mm.

**Figure 2 bioengineering-11-00794-f002:**
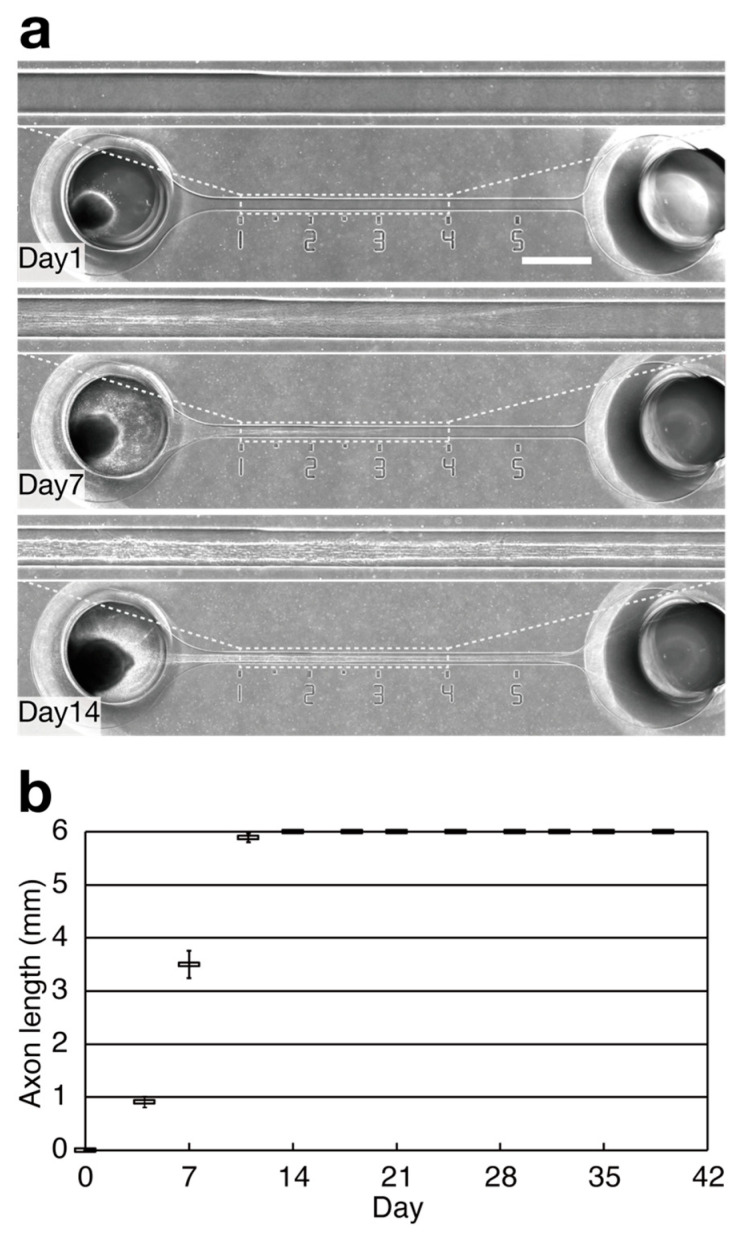
The formation of sensory nerve organoids. (**a**) Axon elongation in the microchannel. The 1–4 mm regions from the entrance of the microchannel are magnified. Scale bar: 1 mm. (**b**) The length of the leading axons in the microchannels. The vertical axis value of 6 mm indicates an axon length of 6 mm or more. The error bars indicate ± the standard error of the mean (SEM), *n* = 12.

**Figure 3 bioengineering-11-00794-f003:**
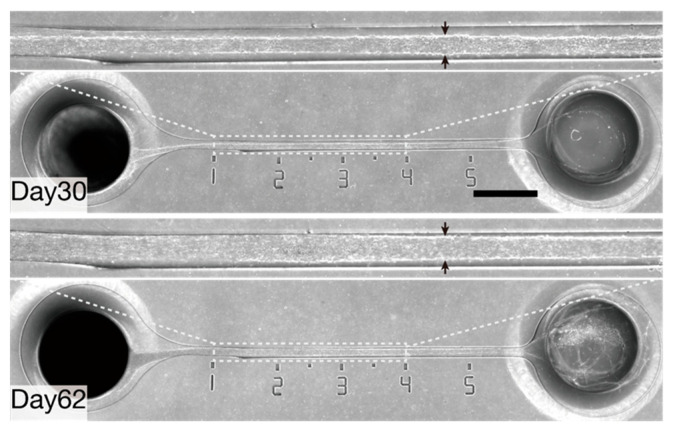
Representative images of a series of long-term cultured sensory nerve organoids. Axon bundle diameters at 3 mm from the entrance of the microchannel are indicated by two arrows. The 1–4 mm regions from the entrance of the microchannel are magnified. Scale bar: 1 mm.

**Figure 4 bioengineering-11-00794-f004:**
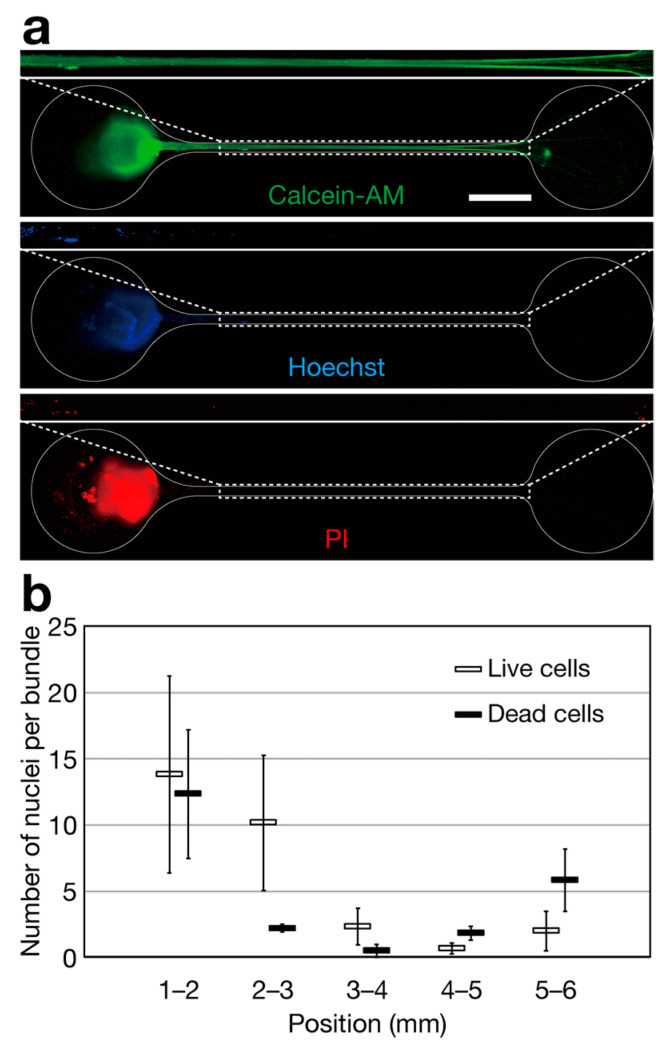
Analysis of live cell contamination in the axon bundles. (**a**) Visualization of the nuclei using fluorescence probes. The 1–6 mm regions from the entrance of the microchannel are magnified. Scale bar: 1 mm. (**b**) The number of nuclei derived from live and dead cells in each region of the axon bundles. The position indicates the distance from the entrance of the microchannels. Organoids cultured for 19 days were used. The error bars indicate ± SEM, *n* = 6.

**Figure 5 bioengineering-11-00794-f005:**
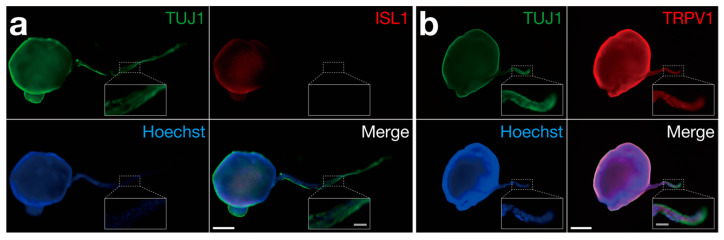
Characterization of neurospheres and axon bundles using fluorescent immunostaining. (**a**,**b**) TUJ1 as a neuron marker and (**a**) ISL1 as a sensory neuron marker. An organoid cultured for 39 days was used. (**b**) TRPV1 as a nociceptor marker. An organoid cultured for 28 days was used. White-colored scale bars: 500 μm. Gray-colored scale bars in close-up panels for axon bundles: 100 μm.

**Figure 6 bioengineering-11-00794-f006:**
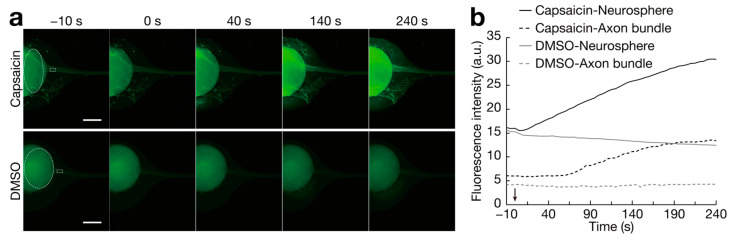
Response of sensory nerve organoids to capsaicin stimulation. (**a**) Fluo 4-AM intensity indicating intracellular calcium ion influx following capsaicin (upper panel) and DMSO (control; lower panel) treatments. Capsaicin solution (100 μM) or DMSO was poured into reservoir of device at 0 s. Scale bars: 1 mm. (**b**) Time course of Fluo 4-AM fluorescent intensity in neurospheres and axon bundles. Timing for pouring 100 μM capsaicin solution or DMSO at 0 s is indicated by an arrow. ROIs for intensity measurements are dotted ellipses for neurospheres and dotted rectangles for axon bundles in (**a**). Representative data of organoids cultured for 32 days were used.

## Data Availability

The data presented in this study are available from the corresponding author on request.

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
