# Peer review of "Formation and Long-Term Culture of hiPSC-Derived Sensory Nerve Organoids Using Microfluidic Devices"

_bioengineering, 2024, doi:10.3390/bioengineering11080794_

Round 1

Reviewer 1 Report

Comments and Suggestions for Authors

1.      What do the authors mean with „induced sensory neurons“ at line 39? If it refers to iPSC-derived neurons that it is already stated right after…

2.      Was the IF in Fig 5 performed for only once? The authors could provide a higher resolution image of the IF or a close-up on the axon bundles to better show the staining.

3.      What is the „n“ of the experiments shown in Figure 6?

4.      Did the authors check if the nerve organoids had the same molecular markers and were still responsive to capsaicin stimulation at later culture time points (e.g. 60 days)?

5.      The authors could perform a flow cytometry or IF analysis to identify which other cell types are present in the organoid that may also explain which other cell types are in migrating/preset in the axon bundles.

Reviewer 2 Report

Comments and Suggestions for Authors

The manuscript provides an interesting study on sensory nerve formation in a microfluidic chip. While overall the study is sound and unique, it appears that the novelty of the chip itself and the type of experiment is questionable given previous publications by the authors already in 2017. It would have been appreciated to also explain to the readers in which way the chip design has been improved or more detailed characterized compared to the earlier publication (Stem Cell Reports v.9(5); 2017 Nov 14 PMC5831012).
